# Evaluation of Anti-Mycobacterial Compounds in a Silkworm Infection Model with *Mycobacteroides abscessus*

**DOI:** 10.3390/molecules25214971

**Published:** 2020-10-27

**Authors:** Kanji Hosoda, Nobuhiro Koyama, Hiroshi Hamamoto, Akiho Yagi, Ryuji Uchida, Akihiko Kanamoto, Hiroshi Tomoda

**Affiliations:** 1Department of Microbial Chemistry, Graduate School of Pharmaceutical Sciences, Kitasato University, Tokyo 108-8641, Japan; hosodak@pharm.kitasato-u.ac.jp (K.H.); koyaman@pharm.kitasato-u.ac.jp (N.K.); 2Medicinal Research Laboratories, School of Pharmacy, Kitasato University, Tokyo 108-8641, Japan; 3Institute of Medical Mycology, Teikyo University, Tokyo 192-0395, Japan; hamamotoh@main.teikyo-u.ac.jp; 4Faculty of Pharmaceutical Sciences, Tohoku Medical and Pharmaceutical University, Sendai 981-8558, Japan; 21752501@is.tohoku-mpu.ac.jp (A.Y.); uchidar@tohoku-mpu.ac.jp (R.U.); 5OP Bio Factory Co. Ltd., Okinawa 904-2234, Japan; akihiko.kanamoto@opbio.com

**Keywords:** silkworm, mycobacteria, nontuberculous mycobacteria, *Mycobacterium avium* complex, *Mycobacteroides abscessus*, antimycobacterial activity, microbial product, lariatin, nosiheptide, ohmyungsamycin, steffimycin, quinomycin

## Abstract

Among four mycobacteria, *Mycobacterium avium*, *M. intracellulare*, *M. bovis* BCG and *Mycobacteroides* (*My.*) *abscessus*, we established a silkworm infection assay with *My. abscessus*. When silkworms (fifth-instar larvae, *n* = 5) were infected through the hemolymph with *My. abscessus* (7.5 × 10^7^ CFU/larva) and bred at 37 °C, they all died around 40 h after injection. Under the conditions, clarithromycin and amikacin, clinically used antimicrobial agents, exhibited therapeutic effects in a dose-dependent manner. Furthermore, five kinds of microbial compounds, lariatin A, nosiheptide, ohmyungsamycins A and B, quinomycin and steffimycin, screened in an in vitro assay to observe anti-*My. abscessus* activity from 400 microbial products were evaluated in this silkworm infection assay. Lariatin A and nosiheptide exhibited therapeutic efficacy. The silkworm infection model with *My. abscessus* is useful to screen for therapeutically effective anti-*My. abscessus* antibiotics.

## 1. Introduction

In the process of antibiotic discovery, candidate compounds active against pathogenic microorganisms in an in vitro assay system often have no therapeutic effects in in vivo animal infection models. Therefore, therapeutic efficacies of candidate compounds in an in vivo assay need to be evaluated at the early stage of drug development. However, in vivo evaluation using mice, rats or rabbits is time-consuming and expensive, in addition to having ethical issues. In order to overcome these issues, we established in vivo-mimic infection assays using silkworms (fifth-instar larvae) with methicillin-resistant *Staphylococcus aureus* (MRSA) [1,2,3,4], *Pseudomonas aeruginosa* and *Candida albicans* [5,6,7,8]. Thus, the silkworm infection model has many advantages over the mouse infection model such as fewer ethical issues, lower maintenance costs, less space required to keep animals, less drugs required for evaluations and shorter times for infection experiments [6,7].

Pulmonary diseases caused by non-tuberculous mycobacteria (NTM) are increasing worldwide. Of note, in several areas, including the United States, Canada and Japan, the incidence rate of NTM disease is higher than that of tuberculosis (TB) [9,10,11]. The major causative agents of NTM diseases are *Mycobacterium avium*, *M. intracellulare* (a mixed infection with *M. avium* and *M. intracellulare* is called *Mycobacterium avium* complex (MAC)) and *Mycobacteroides* (*My.*) *abscessus* for more than 90% of the patients with NTM disease. Although clarithromycin (CAM), rifampicin (RFP) and ethambutol (EB) are used for pulmonary NTM infection, their therapeutic effects are limited [12,13]. Therefore, it is currently important to discover new drugs for the treatment of NTM infection. Accordingly, we started to search for new microbial antibiotics active against NTM. In the present study, we first found that lariatins [14], nosiheptide [15,16], ohmyungsamycins [17], steffimycin [18,19] and quinomycin A [20] exhibited in vitro anti-mycobacterial activity against *M. avium*, *M. intracellulare* and *My. abscessus*. As a next step, we investigated the assay conditions for infecting silkworms with *M. avium*, *M. intracellulare*, *M. bovis* and *My. abscessus*. As a result, *My. abscessus* killed all silkworms efficiently and we established a silkworm infection assay with *My. abscessus.* Using this infection model, clinically used mycobacterial agents and screened anti-NTM antibiotics were evaluated.

## 2. Results

### 2.1. Establishment of a Silkworm Infection Assay with My. abscessus

Based on the conditions for the silkworm infection assay with *M. smegmatis* [21], the temperature (37 °C) for breeding silkworms (fifth-instar larvae) and the colony number of the four mycobacteria (*M. avium*, *M. intracellulare*, *M. bovis* and *My. abscessus*) injected into silkworms were investigated. When the four mycobacteria were injected into silkworms at the highest CFU and the silkworms were bred at 37 °C, only *My. abscessus* injection (1.5 × 10^8^ CFU/larva) killed all silkworms (*n* = 5) within 40 h (Figure 1). Silkworms lived much longer or did not die by injection of the other mycobacteria, including a mixture of *M. avium* and *M. intracellulare*. Thus, among the mycobacteria, a silkworm infection assay can be applied for *My. abscessus.*

Next, seven different colony numbers (1.5 × 10^6^ to 1.5 × 10^8^ CFU/larva) of *My. abscessus* were subsequently injected into silkworms and infected silkworms were bred at 37 °C. As shown in Figure 2, all silkworms died in a colony number-dependent manner. The times to kill all silkworms were 38 (1.5 × 10^8^ CFU/larva injection), 43 (7.5 × 10^7^ CFU/larva), 49 (3.7 × 10^7^ CFU/larva), 59 (1.5 × 10^7^ CFU/larva) and 68 h (1.5 × 10^6^ CFU/larva). Based on these results, the conditions for the silkworm infection model with *My. abscessus* were established: injection at 7.5 × 10^7^ CFU/larva and breeding at 37 °C. Under these conditions, all silkworms died after 43 h.

### 2.2. Therapeutic Efficacies of Drugs in the Silkworm Infection Assay with My. abscessus

Firstly, the toxicity of four clinically used antimicrobial agents (clarithromycin (CAM), amikacin (AMK), imipenem (IPM) and ciprofloxacin (CPFX)) to silkworms was tested (50 μg/larva, *n* = 5). CAM, AMK and IPM alone exhibited no effect on silkworms. However, CPFX caused silkworm death within 50 h. Then, the four drugs were evaluated in the silkworm infection assay (*n* = 5) with *My. abscessus*. When CAM, AMK and IPM were injected, silkworms survived in a dose-dependent manner (Figure 3). They exhibited no toxic effects on silkworms at 50 μg/larva. On the other hand, CPFX exerted therapeutic effects at 3.12 and 12.5 μg/larva, but CPFX was toxic to silkworms at 50 μg/larva. Among them, CAM exhibited the strongest therapeutic activity (50% effective dose (ED_50_), 0.22 μg/larva), followed by AMK (ED_50_, 1.48 μg/larva). The ED_50_ values of the four antimicrobial agents against *My. abscessus* are summarized in Table 2.

### 2.3. Screening for Anti-NTM Compounds from a Microbial Product Library

Our microbial product library consisting of approximately 400 pure compounds was screened for compounds active against all three NTM, *M. avium*, *M. intracellulare* and *My. abscessus,* using the liquid microdilution method [16]. As a result, six microbial compounds (lariatin A (**1**), nosiheptide (**2**), ohmyungsamycin A (**3**), ohmyungsamycin B (**4**), quinomycin A (**5**) and steffimycin (**6**)) (Figure 4) demonstrated potent anti-NTM activity. The minimum inhibitory concentration (MIC) values are summarized in Table 1. Among them, quinomycin A (**5**) was the most effective against *My. abscessus* (MIC, 0.01 μg/mL), followed by lariatin A (**1**) (MIC, 1.56 μg/mL) and nosiheptide (**2**) (MIC, 1.56 μg/mL). The MICs of CAM, AMK, IPM and CPFX against *My. abscessus* are also shown in Table 1 for comparative purposes. We demonstrated that compounds **1**–**6** exhibited anti-*My. abscessus* activity.

### 2.4. Therapeutic Efficacies of Microbial Compounds in the Silkworm Infection Assay with My. abscessus

The study compounds were evaluated in the silkworm infection assay. The survival rate of infected silkworms (*n* = 5) after injection of the study compounds at a dose of 50 μg/larva is shown in Figure 5. Under the conditions in which all infected silkworms (control) died within 48 h, all of the study compounds except **5** prolonged the survival of infected silkworms. In contrast, **5**, which had the most potent anti-*My. abscessus* activity in vitro, markedly reduced the survival, probably due to its marked toxic effects on silkworms. Next, the dose dependency of **1**–**4** was investigated (Figure 6). Compounds **1** and **2** prolonged the survival in a dose-dependent manner, whereas **3** and **4** had subtle effects with dose dependency. The ED_50_ values of these compounds against *My. abscessus* are summarized in Table 2. Compounds **1** and **2** exerted therapeutic effects with respective ED_50_ values of 8.84 and 14.6 μg/larva in the silkworm infection assay. All of the study compounds except **5** (50 μg/larva) did not exhibit toxicity against silkworms for at least 48 h.

## 3. Discussion

In the present study, an in vivo-mimic silkworm infection model with four mycobacteria, *M. avium*, *M. intracellulare, M. bovis* and *My. abscessus*, was evaluated. Although the breeding temperature of silkworms (27 or 37 °C) and the colony number of the mycobacteria for infection were set based on the previous study of a silkworm infection assay with *M. smegmatis* [21], the silkworms infected with *M. avium, M. intracellulare* and *M. bovis* did not die within 70 h. Only the silkworms infected with *My. abscessus* died around 40 h after infection. *M. smegmatis* and *My. abscessus,* rapidly growing mycobacteria, may grow in silkworms, leading to death, whereas *M. avium*, *M. intracellulare* and *M. bovis,* slowly growing mycobacteria, have no ability to kill silkworms. The breeding temperature should be 37 °C (infected silkworms did not die at 27 °C), the best condition for mycobacterial growth, suggesting that the growth speed of mycobacteria is important for this infection assay (Figure 1).

Four clinically used anti-mycobacterial agents were evaluated in this silkworm assay (Figure 3). The order of potency in the in vitro anti-*My. abscessus* assay (MIC in Table 2) was CAM > CPFX > AMK = IPM, whereas that in the silkworm assay (ED_50_ in Table 2) was CAM > AMK > CPFX > IPM. The orders in both assays are not exactly the same, but they have a similar tendency except for CPFX. CPFX did not exhibit therapeutic efficacy at the highest dose due to its toxicity or insecticidal activity. Sekimizu and colleagues demonstrated that the therapeutic efficacies of clinically used drugs in a silkworm infection assay are consistent with those in a mouse infection assay [7]. Accordingly, the present study suggested the clinical importance of CAM and AMK for the treatment of *My. abscessus* patients. *My. abscessus* was reported to have inducible resistance to CAM by the *erm*(41) gene [22], but it may be difficult to observe such inducible resistance in this silkworm assay because of the short evaluation time (within 70 h). There was no preclinical mouse model with *My. abscessus*. The zebrafish model was generally utilized to evaluate in vivo efficacy until recently [23]. In 2020, Riva et al. reported a new model with immunocompetent mice [24,25]. In this study, we demonstrated the usefulness and effectiveness of silkworm for the first time. Therefore, we consider this silkworm infection model with *My. abscessus* to be applicable to evaluate the in vivo efficacy of candidate compounds as anti-*My. abscessus* agents.

From our microbial product library (400 compounds), six compounds exhibited in vitro anti-NTM activity and they were evaluated in this silkworm infection assay with *My. abscessus*. Lariatin A (**1**) (produced by *Rhodococcus jostii* K01-B0171), originally discovered as a selective anti-*M. smegmatis* antibiotic, also exhibited anti-TB activity [14]. This unique lasso peptide was reported to be effective in the silkworm infection assay with *M. smegmatis* [21]. In the present study, we demonstrated that **1** was also active against *My. abscessus* in vitro and in this silkworm infection assay. Nosiheptide was discovered as an anti-Gram-positive antibiotic in 1977 [15]. Recently, we reported that nosiheptide has potent in vitro anti-mycobacterial activity against MAC and *M. smegmais*. Of note, it was not therapeutically effective in the silkworm infection assay with *M. smegmatis* [16], but the antibiotic was active in the silkworm infection assay with *My. abscessus* (Figure 6b). In addition, nosiheptide was reported to have therapeutic effects in a mouse infection assay with *Staphylococcus aureus* [26]. Ohmyungsamycins were discovered as anti-TB antibiotics in 2013 [17,27], exhibiting weak in vitro and in vivo activity against *My. abscessus.* Steffimycin, discovered as an anti-cancer antibiotic in 1977 [18,19,28], demonstrated weak in vitro activity against *My. abscessus*, but no therapeutic effects in the silkworm infection assay with *My. abscessus*. Quinomycin A, reported as an anti-Gram-positive and anti-cancer antibiotic in 1961 [20,29], exhibited the strongest in vitro activity against *My. abscessus,* but all silkworms treated with quinomycin A died earlier than infected silkworms (control). Thus, the silkworm infection assay can evaluate not only therapeutic efficacy, but also toxicity. In summary, we demonstrated that microbial compounds **1**–**6** exhibited anti-*My. abscessus* activity, and that **1** and **2** retained therapeutic efficacy in the silkworm infection assay with *My. abscessus*. We consider this silkworm infection model to be applicable to the evaluation of the in vivo effectiveness of candidate compounds.

Hamamoto et al. previously reported that the ED_50_/MIC value of a compound is an index of drug potential, and the ratio is typically below 10 for clinically useful antibiotics [7]. As shown in Table 2, the ratios of **1** and **2** were below 10, suggesting that they are potential anti-*My. abscessus* drugs.

In conclusion, we established an in vivo-mimic silkworm infection assay with *My. abscessus.* The therapeutic efficacies of clinically used anti-NTM drugs and microbial compounds were evaluated in this silkworm infection assay, suggesting their potential in vivo efficacies. As chemotherapeutic drugs for the treatment of *My. abscessus* patients are limited, this silkworm infection model will be valuable to select practically effective anti-*My. abscessus* drug candidates.

## 4. Materials and Methods

### 4.1. Materials

Lariatin A, nosiheptide, ohmyungsamycins, steffimycin and quinomycin A were purified from a culture broth of actinomycetes in our laboratory. Clarithromycin (CAM), amikacin (AMK), imipenem (IPM) and ciprofloxacin (CPFX) were purchased from Wako Pure Chemical Industries (Osaka, Japan). Middlebrook 7H9 broth (Becton, Dickinson and Company, Franklin Lakes, NJ, USA) containing 0.05% Tween 80 (Tokyo Chemical Industries: Tokyo, Japan) and 10% albumin dextrose catalase (ADC) enrichment (5% bovine serum albumin, Sigma Aldrich (MO, USA); 2% glucose, Wako Pure Chemical Industries (Osaka, Japan); 0.85% NaCl, Wako Pure Chemical Industries) was used for the cultivation of mycobacteria. Third-molting larvae stage silkworms, Bombyx mori (Hu•yo  ×  Tukuba•Ne), were purchased from Ehime Sanshu (Ehime, Japan). Silk Mate 2S, an artificial diet containing antibiotics, was purchased from Nosan Corporation (Kanagawa, Japan). The following mycobacterial strains were used in this study: *M. avium* JCM15430, *M. intracellulare* JCM6384, *M. bovis* BCG Pasteur and *My. abscessus* ATCC19977. The *M. bovis* BCG Pasteur used was a laboratory strain. *M. avium* JCM15430 and *M. intracellulare* JCM6384 were purchased from the Riken BioResource Research Center (Ibaraki, Japan). *My. abscessus* ATCC19977 was purchased from the American Type Culture Collection (VA, USA).

### 4.2. Preparation of Mycobacterial Suspension

*M. avium* JCM15430, *M. intracellulare* JCM6384, *M. bovis* BCG Pasteur and *My. abscessus* ATCC19977 were stored in 20% glycerol at −80 °C. The frozen stock cultures of *M. avium, M. intracellulare*, *M. bovis* and *My. abscessus* (200, 120, 500 and 200 μL) were inoculated in Middlebrook 7H9 broth (6 mL) in a T-25 flask (Corning, Corning, NY, USA), and cultured under static conditions at 37 °C for 96, 72, 96 and 48 h, respectively (>1 × 10^8^ CFU/mL).

### 4.3. MIC Values in the Liquid Microdilution Assay

Anti-mycobacterial activities against these four strains were evaluated according to a previously established liquid microdilution method [16].

*M. avium, M. intracellulare*, *M. bovis* and *My. abscessus* suspensions were adjusted to 4.0 × 10^6^–1×10^7^ CFU/mL in Middlebrook 7H9 broth containing 0.05% Tween 80 and 10% ADC enrichment. The suspension (95 μL) was added to each well of a 96-well microplate (Corning) with or without the test drugs (5 μL in MeOH or water) and incubated at 37 °C for 72–120 h. MTT reagent (5.5 mg/mL MTT, 5 μL) was added to each well and the cells were incubated for 16 h. After cells were lysed with lysis buffer (40% *N,N*-dimethylformamide, Nacalai Tesque, Kyoto, Japan; 20% SDS, Wako Pure Chemical Industries; 2% CH_3_COOH, Kanto Chemical, Tokyo, Japan; 95 μL), the absorbance of the lysate was measured at 570 nm using an absorption spectrometer. The MIC value was defined as the lowest drug concentration that resulted in 90% growth inhibition of *M. avium, M. intracellulare*, *M. bovis* and *My. abscessus.*

### 4.4. Silkworm Infection Assay with Mycobacteria Spp.

Hatched silkworm larvae were raised by feeding an artificial diet containing antibiotics (Silk Mate 2S) in an incubator at 27 °C until the fourth molting stage. On the first day of fifth-instar larvae, silkworms (*n* = 5) were fed Silk Mate 2S. On the second day, mycobacterial suspensions (6.0 × 10^7^ to 7.0 × 10^8^ CFU/larva) were injected into the hemolymph through the dorsal surface of the silkworm using a disposable 1-mL syringe with a 27-G needle (TERUMO, Tokyo, Japan). After injection, the number of silkworms that survived was counted at the indicated time until 80 h. The data are plotted according to the Kaplan–Meier method [30].

### 4.5. Silkworm Infection Assay with My. abscessus

Seven different cell numbers (1.5 × 10^6^ to 1.5 × 10^8^ CFU/larva) of *My. abscessus* suspensions were subsequently injected into the hemolymph through the dorsal surface of the silkworm (2.0 g, *n* = 5) using a disposable 1-mL syringe with a 27-G needle, and the number of silkworms that survived was counted at the indicated time until 80 h. The data are plotted according to the Kaplan–Meier method [30]

### 4.6. ED_50_ Values in the Silkworm Infection Assay with My. abscessus

A *My. abscessus* ATCC19977 suspension (7.5 × 10^7^ CFU/larva in 50 μL Middlebrook 7H9 broth) was injected into the hemolymph of silkworm larvae (2.0 g, *n* = 5), followed by the injection of anti-*My. abscessus* drugs or microbial compounds (50 μL in water or 10% DMSO) within 30 min. Silkworms were maintained at 37 °C. The dose of a sample leading to a 50% survival rate (ED_50_) was calculated at the time when all *My. abscessus*-infected silkworms without sample injection died (around 43 h), according to a previous method [7,21,31].

## Figures and Tables

**Figure 1 molecules-25-04971-f001:**
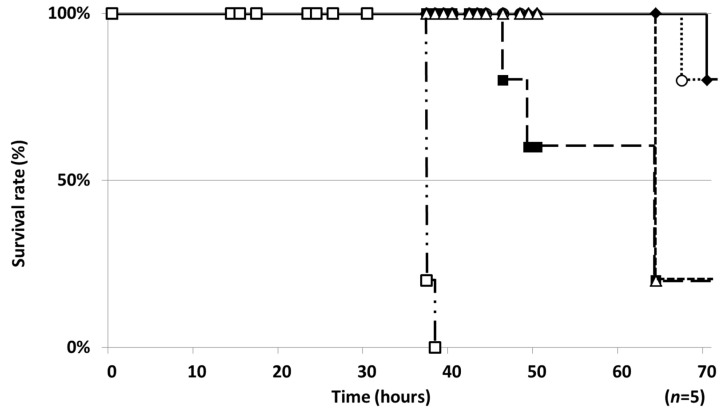
Silkworm-killing ability of four mycobacteria. A suspension of four mycobacteria strains was diluted to the indicated cell number and injected into the silkworm hemolymph. Infected silkworms were incubated at 37 °C. The number of surviving silkworms was counted 70 h after the injection. ◆: *M. avium* JCM15430 (6.0 × 10^7^ CFU/larva); ■: *M. intracellulare* JCM6384 (7.0 × 10^8^ CFU/larva); △: *M. avium* + *M. intracellulare* (1.8 × 10^8^ + 3.5 × 10^8^ CFU/larva); ○: *M. bovis* BCG Pasteur (5.0 × 10^8^ CFU/larva); □: *My. abscessus* ATCC19977 (1.5 × 10^8^ CFU/larva). Experiments were performed three times and reproducible data were observed.

**Figure 2 molecules-25-04971-f002:**
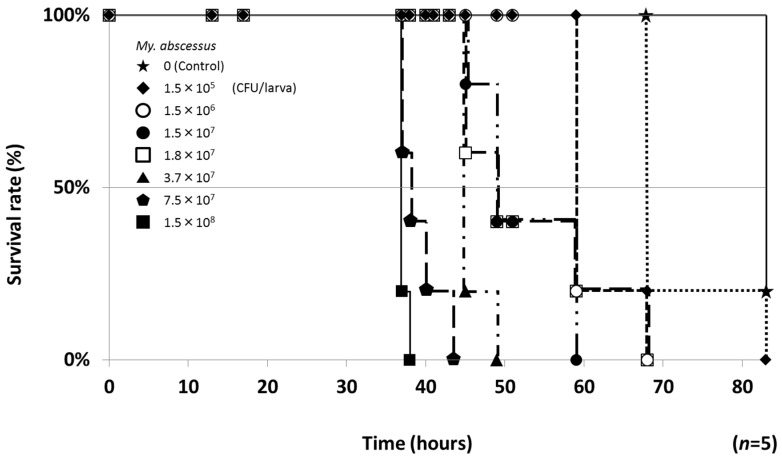
Silkworm-killing ability of *My. abscessus*. A suspension of *My. abscessus* ATCC19977 strain was diluted to the indicated cell number and injected into the silkworm hemolymph. Infected silkworms were incubated at 37 °C. The number of surviving silkworms was counted until 83 h after the injection. Experiments were performed two times and reproducible data were observed.

**Figure 3 molecules-25-04971-f003:**
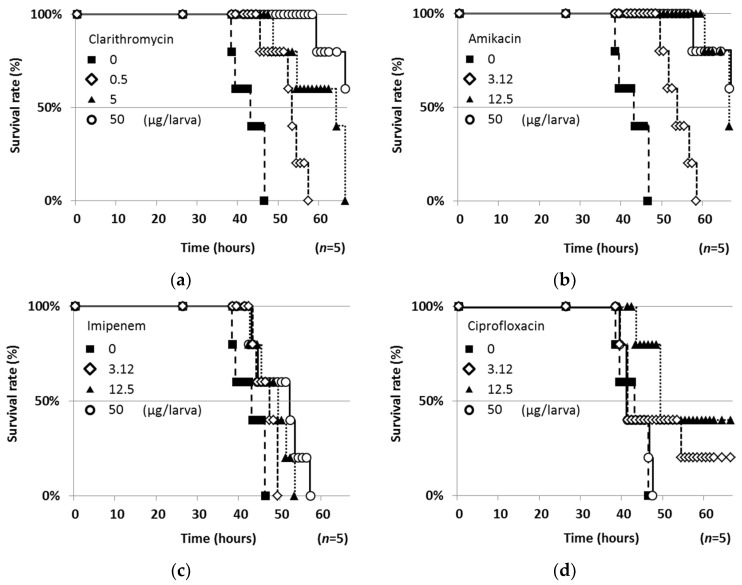
Therapeutic effects of anti-*My. abscessus* drugs in the silkworm infection assay with *My. abscessus*. (**a**) Clarithromycin, (**b**) amikacin, (**c**) imipenem and (**d**) ciprofloxacin.

**Figure 4 molecules-25-04971-f004:**
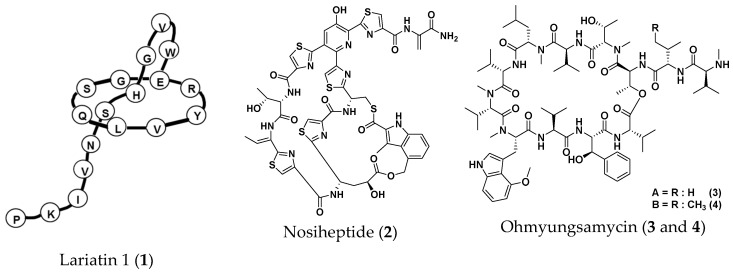
Structures of microbial anti-non-tuberculous mycobacteria (NTM) compounds.

**Figure 5 molecules-25-04971-f005:**
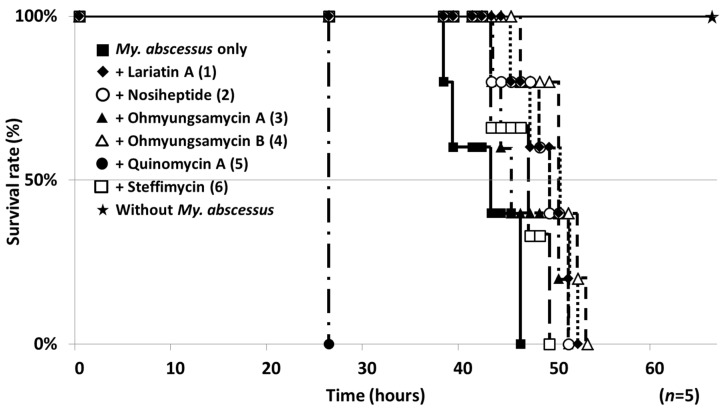
Assay results of microbial anti-NTM compounds in the silkworm infection assay with *My. abscessus*. The silkworms were injected with *My. abscessus* (7.5 × 10^7^ CFU/larva) and the test compound at 50 μg/larva.

**Figure 6 molecules-25-04971-f006:**
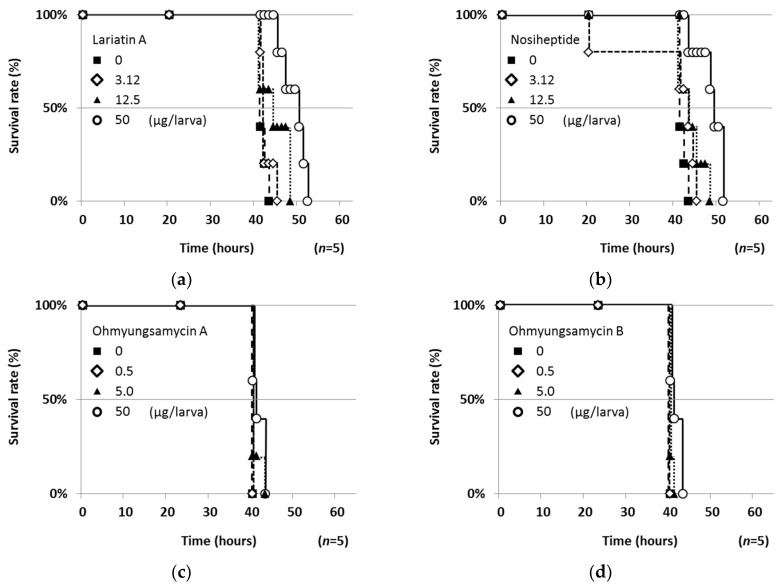
Therapeutic effects of microbial anti-NTM compounds in the silkworm infection assay with *My. abscessus*. (**a**) Lariatin A (**1**), (**b**) nosiheptide (**2**), (**c**) ohmyungsamycin A (**3**) and (**d**) ohmyungsamycin B (**4**). Experiments were performed twice to confirm reproducibility.

**Table 1 molecules-25-04971-t001:** MIC values of anti-NTM agents against four mycobacteria.

Test Microorganism	MIC (μg/mL) *									
	1	2	3	4	5	6	CAM	AMK	IPM	CPFX
*M. avium* JCM15430	0.78	0.02	0.19	0.19	<0.01	1.56	0.19	3.12	>50	0.78
*M. intracellulare* JCM6384	1.56	0.02	0.10	0.10	<0.01	0.39	0.02	0.78	1.56	0.09
*M. bovis* BCG Pasteur	0.78	0.01	N.T.	N.T.	N.T.	1.56	0.12	0.78	>50	0.09
*My. abscessus* ATCC19977	1.56	1.56	12.5	12.5	0.01	25.0	0.39	12.5	12.5	1.56

**1**: lariatin A, **2**: nosiheptide, **3**: ohmyungsamycin A, **4**: ohmyungsamycin B, **5**: quinomycin A, **6**: steffimycin, N.T.: not tested. * minimum inhibitory concentration.

**Table 2 molecules-25-04971-t002:** MIC and ED_50_ values of anti-mycobacterial agents against *My. abscessus.*

Test Compound	ED_50_ (μg/larva) ^1^	MIC (μg/mL)	ED_50_/MIC
Lariatin A (**1**)	8.84	1.56	5.67
Nosiheptide (**2**)	14.6	1.56	9.36
Ohmyungsamycin A (**3**)	30.0	12.5	2.40
Ohmyungsamycin B (**4**)	30.0	12.5	2.40
Quinomycin A (**5**)	>50	25.0	-
Steffimycin (**6**)	>50	0.01	-
CAM	0.22	0.39	0.56
AMK	1.48	12.5	0.12
IPM	7.82	12.5	0.63
CPFX	5.47 *	1.56	3.51 *

1: 50% effective dose. *: all silkworms died at 50 μg/larva.

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
