# Peer review of "Evaluation of Anti-Mycobacterial Compounds in a Silkworm Infection Model with Mycobacteroides abscessus"

_molecules, 2020, doi:10.3390/molecules25214971_

Round 1

Reviewer 1 Report

Hosoda et al. developed a silkworm model that was used to assess the potential therapeutic activity of several compounds against Mycobacterium abscessus. Overall, I believe the manuscript is relatively well-written, and the results of the study will be of interest to potential readers; however, I have several comments that may improve the presentation of the manuscript.

  • Can the authors explain why they chose the commercial strains that were used in the initial experiment to determine if the different Mycobacteria species were capable of killing the silk worms? Is the pathogenicity of the strains similar to modern clinical isolates?
  • Similarly, the in vitro MIC values and the silkworm studies utilized a single strain of M. abscessus. Is it possible to assess the in vitro activity of the study compounds against additional M. abscessus isolates to provide a better idea of how the study results extrapolate to other isolates? If not, the authors should address in the Discussion that a limitation of the study is that the conclusions from the present study are based on a single strain.
  • Is there an explanation for why so many of the survival plots contain precipitous drops where most of the silkworms died simultaneously? For example, in Figure 5 did all 5 of the silkworms that received quinomycin A die at ~26 hours? It may be easier to conceptualize what happened to the worms if the authors disclose which time points were used to assess the survival of the worms (Line 232 in the Methods simply states that surviving worms were counted at the “indicated time”).
  • Visually, Figure 3 and Figure 6 may be easier to interpret if the authors consider alternating between filled and hallow symbols. The filled symbols are small and are difficult to distinguish from one another. In contrast, Figure 2 contains a few empty symbols. If the authors were to alternate filled and empty symbols in Figure 3 and Figure 6 I think it will be easier for the reader to correctly identify the correct symbol from the legend. The legend itself may also be improved by including the unique lines used for each drug/compound amount into the legend (for example, a square with a solid line versus a circle with a dotted line).
  • I also think having the drug/compound names listed somewhere in each panel for Figure 3 and Figure 6 (either in the legends or above the plots) will make it easier for the reader to interpret the figure.
  • In lines 99 – 107, the authors introduce numbers that are used to refer to each of the study compounds; however, the inconsistent use of the numbers in the rest of the manuscript makes the interpretation of the study cumbersome. For example, Table 1 only uses numbers to refer to the compounds, so the reader is forced to return to lines 99 – 107 to interpret Table 1. Then in Figure 5 and Figure 6, only the compound names are used and none of the numbers are included, whereas Table 2 uses both the compound names and the numbers. The description of the results relating to Figure 6 is completed only using the numbers, so trying to read the description of the results while comparing the corresponding figure is very tedious. I personally think the benefit of not having to repeat the compound names is outweighed by confusion introduced by the numbers. Minimally, I think the authors should include the full compound names as a footnote to Table 1, and I believe the authors should write out the compound names in the rest of the manuscript or use abbreviations that clearly correspond to each compound. For example, instead of referring to “antibiotics 1-6” in lines 114 – 115, the authors can simply state that the “study compounds” were evaluated, and instead of stating that “1, 2, 3, 4, and 6 prolonged the survival” of worms in line 117, the authors can simply state that all of the study compounds except quinomycin A prolonged the survival of the worms.
  • I am curious how the authors determined whether a study compound provided a substantial improvement in the survival of the worms. There aren’t any statistics used in the manuscript, and some of the survival benefits are on the scale of a few hours. It might be nice to have a statement in the Methods that a survival increase of X number of hours for X worms was considered a relevant survival benefit or something of that nature.

Author Response

Thank you for your suggestion.

We agree with you and have incorporated this suggestion throughout our paper.

Point 1:Can the authors explain why they chose the commercial strains that were used in the initial experiment to determine if the different Mycobacteria species were capable of killing the silk worms? Is the pathogenicity of the strains similar to modern clinical isolates?

Similarly, the in vitro MIC values and the silkworm studies utilized a single strain of M. abscessus. Is it possible to assess the in vitro activity of the study compounds against additional M. abscessus isolates to provide a better idea of how the study results extrapolate to other isolates? If not, the authors should address in the Discussion that a limitation of the study is that the conclusions from the present study are based on a single strain.

Response 1:Thank you for your comment. We used commercially available mycobacteria in this study, because they have been used in many papers and they are originally clinical isolates. Therefore, we believe that our experiments are reasonable. In fact, anti-mycobacterial activities of known drugs against these strains are within the acceptable levels. Although the silkworm infection assay may be limited to this strain, the strain has been used as the standard My. abscessus and our results are important for many researchers.

Point 2: Is there an explanation for why so many of the survival plots contain precipitous drops where most of the silkworms died simultaneously? For example, in Figure 5 did all 5 of the silkworms that received quinomycin A die at ~26 hours? It may be easier to conceptualize what happened to the worms if the authors disclose which time points were used to assess the survival of the worms (Line 232 in the Methods simply states that surviving worms were counted at the “indicated time”).

I am curious how the authors determined whether a study compound provided a substantial improvement in the survival of the worms. There aren’t any statistics used in the manuscript, and some of the survival benefits are on the scale of a few hours. It might be nice to have a statement in the Methods that a survival increase of X number of hours for X worms was considered a relevant survival benefit or something of that nature.

Response 2: Thank you for your comment. We observed 5 silkworms as a group for an assay. The survival rates are plotted by the typical “Kaplan-Meier” method. If we use 100 silkworms for an assay, the curves are plotted as the reviewer’s image. The “Kaplan-Meier” method was cited in Materials and Methods (lines 235-236 and lines 241-242).

Point 3: Visually, Figure 3 and Figure 6 may be easier to interpret if the authors consider alternating between filled and hallow symbols. The filled symbols are small and are difficult to distinguish from one another. In contrast, Figure 2 contains a few empty symbols. If the authors were to alternate filled and empty symbols in Figure 3 and Figure 6 I think it will be easier for the reader to correctly identify the correct symbol from the legend. The legend itself may also be improved by including the unique lines used for each drug/compound amount into the legend (for example, a square with a solid line versus a circle with a dotted line).

I also think having the drug/compound names listed somewhere in each panel for Figure 3 and Figure 6 (either in the legends or above the plots) will make it easier for the reader to interpret the figure.

Response 3: According to your suggestion, we revised Figure 2, 3 and 6 for readers to interpret more easily.

Point 4: In lines 99 – 107, the authors introduce numbers that are used to refer to each of the study compounds; however, the inconsistent use of the numbers in the rest of the manuscript makes the interpretation of the study cumbersome. For example, Table 1 only uses numbers to refer to the compounds, so the reader is forced to return to lines 99 – 107 to interpret Table 1. Then in Figure 5 and Figure 6, only the compound names are used and none of the numbers are included, whereas Table 2 uses both the compound names and the numbers. The description of the results relating to Figure 6 is completed only using the numbers, so trying to read the description of the results while comparing the corresponding figure is very tedious. I personally think the benefit of not having to repeat the compound names is outweighed by confusion introduced by the numbers. Minimally, I think the authors should include the full compound names as a footnote to Table 1, and I believe the authors should write out the compound names in the rest of the manuscript or use abbreviations that clearly correspond to each compound. For example, instead of referring to “antibiotics 1-6” in lines 114 – 115, the authors can simply state that the “study compounds” were evaluated, and instead of stating that “1, 2, 3, 4, and 6 prolonged the survival” of worms in line 117, the authors can simply state that all of the study compounds except quinomycin A prolonged the survival of the worms.

Response 4: Thank you for your thoughtful suggestion. We added the full compound names as a footnote to Table 1 and fixed line 116-119.

Reviewer 2 Report

The author descript a silkworm infection assay with My. abscessus. Four clinically used antimicrobial agents (clarithromycin (CAM), amikacin (AMK), imipenem (IPM) and ciprofloxacin (CPFX)) as the positive control were evaluated in the silkworm infection assay. The clarithromycin and amikacin showed efficacy. the author also screened five kinds of microbial compounds, lariatin A, nosiheptide, ohmyungsamycins A and B, quinomycin and steffimycin to observe anti-My. abscessus activity in vitro from 400 microbial products. then they evaluated them in this silkworm infection assay. Lariatin A and nosiheptide exhibited therapeutic efficacy. the manuscript can be published after revising. some problems should be answered as follows: 1)From the Figure 2, can the author explain the survival rate for the silkworm in the group without My. abscessus. Why did the author choose 7.5×107 CFU/larva as the infection dose? there is no obvious different in the 1.5×107 CFU/larva. 2) The author choose the 37oC as the breeding temperature but no according experiment. The author should added the temperature changing experiments. 2) How did the author determined the feeding dose of these drugs on silkworm infection model? 3) The toxic effects of clarithromycin (CAM), amikacin (AMK), imipenem (IPM) and ciprofloxacin (CPFX) should be evaluate on the silkworm model without My. abscessus infection. 4) the image resolution is too low to view for Figure 4.

Author Response

Thank you for your suggestion.

We agree with you and have incorporated this suggestion throughout our paper.

Point 1: From the Figure 2, can the author explain the survival rate for the silkworm in the group without My. abscessus. Why did the author choose 7.5×107 CFU/larva as the infection dose? there is no obvious different in the 1.5×107 CFU/larva.

Response 1: Thank you for your comment. The silkworm in the group without My. abscessus is the control group (please see Figure 2: no My. abscessus). Since the silkworms after injection are not fed, they will starve to death after 70~80 hours. Therefore, to exclude such disturbance (death from starvation), we chose 7.5×107 CFU/larva as the infection dose.

Point 2: The author choose the 37oC as the breeding temperature but no according experiment. The author should added the temperature changing experiments.

How did the author determined the feeding dose of these drugs on silkworm infection model?

Response 2: According to the reviewer’s request, we added the temperature information at line 138 and line 144.

Point 3: The toxic effects of clarithromycin (CAM), amikacin (AMK), imipenem (IPM) and ciprofloxacin (CPFX) should be evaluate on the silkworm model without My. abscessus infection. 4) the image resolution is too low to view for Figure 4.

Response 3: We added the sentence (line 86-89) “Firstly, toxicity of four clinically used antimicrobial agents (clarithromycin (CAM), amikacin (AMK), imipenem (IPM) and ciprofloxacin (CPFX)) to silkworms were tested (50 μg/larva, n=5). CAM, AMK and IPM alone exhibited no effect on silkworms. But CPFX caused silkworm death within 50 h.”.

Point 4: The image resolution is too low to view for Figure 4

Response 4: We fixed it.

Round 2

Reviewer 2 Report

The manuscript can be published after typos and grammer checking.